# *Capparis spinosa* L. Cenopopulation and Biogeochemistry in South Uzbekistan

**DOI:** 10.3390/plants11131628

**Published:** 2022-06-21

**Authors:** Murodjon Isagaliev, Evgeny Abakumov, Avazbek Turdaliev, Muzaffar Obidov, Mavlonjon Khaydarov, Khusnida Abdukhakimova, Tokhirjon Shermatov, Iskandar Musaev

**Affiliations:** 1Department of Soil Science, Zootechnics and Agronomy, Fergana State University, Fergana 150100, Uzbekistan; murodjon-isa@mail.ru (M.I.); avazbek1002@mail.ru (A.T.); muzaffar_bio@mail.ru (M.O.); mav.tichal@gmail.com (M.K.); khusnida83@mail.ru (K.A.); namvilyerloyiha.uz@gmail.com (T.S.); iskandarmusayev255@gmail.com (I.M.); 2Department of Applied Ecology, Faculty of Biology, Saint Petersburg State University, 16 Line of VO, 199178 Saint-Petersburg, Russia; 3All Russia Institute of Agricultural Microbiology, Podbelsky Chausse 3, Pushkin, 199178 Saint-Petersburg, Russia

**Keywords:** medicinal plants, *Capparis spinosa* L., Calcisols, trace elements, cenopopulation analysis, grass, stages of plant development

## Abstract

The article provides an analysis of the cenopopulation and tissues element composition of the medicinal caper plant *Capparis spinosa* L. distributed on Calcisols formed on eroded alluvial-proluvial gravel textured rocks in the south of the Fergana Valley (Uzbekistan, Central Asia). The predominance of immature plants in the cenopopulation was detected in the Arsif hills massive, and quantitative indicators of micronutrients in the vegetative and generative organs of *C. spinosa* L. were determined. The study of biomorphological characteristics of the plant during the growing season (April-October) was carried out in the identified 10 observational experimental field populations. The cenopopulation dynamics and plant development patterns of *Capparis spinosa* L. were characterized for environmental conditions of south Uzbekistan for the first time. Soil, plant element analysis was performed by neutron-activation method. In this case, the samples were irradiated in a nuclear reactor with a neutron flux of 5 × 10^13^ neutrons/cm^2^ s, and their quantities were determined in accordance with the half-life of chemical elements. It has also been compared with research materials conducted by world scientists on the importance and pharmacological properties of botanicals in medicine and the food industry, as well as their botanical characteristics. The plant can serve to conserve soil resources, as it prevents water and wind erosion of dense clay soils in the dry subtropical climate of Central Fergana and could be considered an effective agent of destroyed soils remediation. The development of this plant will contribute to the diversification of agriculture in Uzbekistan (Central Asia) and the development of the food industry and pharmacology.

## 1. Introduction

Mankind has been using various herbs for thousands of years to find a cure for food support, and people have been particularly well aware of the properties of medicinal plants and have been able to use them effectively. Unfortunately, nowadays, natural medicinal plants have been virtually non-existent. If one looks at the countries of the world, the propagation and processing of medicinal plants are widely developed in countries such as China, India, Canada, and the United States. Trade-in medicinal plants is growing in volume and in exports. It is estimated that the global trade in medicinal plants is US$800 million per year [1]. The annual turnover of medicinal plants and processed products is $3 trillion 26 billion dollars. China alone produces 700,000 tons of medicinal plants a year, of which 122,000 tons of raw materials worth $822 million are exported and $50 billion is traded through processing [2]. Despite the high potential of the industry in Uzbekistan, the existing opportunities for planting are not used enough. In 2019, the country exported 19,000 tons of medicinal plants, finished and semi-finished or raw materials worth $48 million. Today, 93 enterprises produce medicines from 89 types of medicinal plants. Only 7% of the volume of natural medicines in total consumption is accounted for by local manufacturers [2]. The results of the analysis show that the protection of natural resources of medicinal plants, their rational use, the establishment of plantations, reproduction of competitive species in domestic and foreign markets, the introduction of new species of medicinal plants suitable for soil climatic conditions, creation of value chain through primary and deep processing shows that the work being done in this area is lagging behind the requirements of the time [2,3]. The role of medical plants in ensuring public health becomes more and more important [4].

One of the most pressing issues of today is the development of technology for the cultivation and cultivation of natural flora and medicinal plant species belonging to foreign flora, targeted research to obtain environmentally friendly products, introduced into medical practice. One such naturally occurring medicinal plant species is the *C. spinosa* L. plant.

Scientific sources and international publications on the Caper (*C. spinosa* L.), a drought-tolerant plant belonging to the genus Capparis of the family Capparidaceae is mainly distributed in arid and semi-arid regions of the tropical and subtropical environments. The plant, as a potential source of valuable nutrients such as vitamins (especially vitamin C, digestible protein, reducing sugars and essential minerals are valued for human food [5]. This plant is well known for the edible flower buds and the fruit (caper berries), both of which are frequently consumed as pickled. The fruit from this species is used to garnish pizza and also added to salads, sauces, and jams [6]. The caper is interesting for plant science due to the specificity of its micropropagation in wildlife and agricultural environments [7]. The caper plant has been used by humans from very ancient times [8]. The present review summarizes information concerning *C. spinosa* L. including agronomic performance, botanical description, taxonomical approaches, traditional pharmacological uses, phytochemical evaluation, and genetic studies [9]. Commercial capers are immature flower buds that can be pickled in salt or vinegar and used as an appetizer or condiment [10]. Hence, capers are included in hundreds of recipes due to their sharp piquant flavor owed to a complex organoleptic profile [11] and are used as a seasoning to add pungency to sauces (e.g., tartare, remoulade, ravigote, etc.) dressings and salads (e.g., caponata, a cold eggplant salad with olives and capers), cold dishes and sauces served with salmon, herring, pasta and pizzas, cheeses, lamb, mutton, pork and chicken preparations [12]. Unripe fruits called caper berries are also pickled and used as spices and condiments [13]. Food industries also use extracts from caper buds and ripened fruits as flavor agents [14] abound. The seeds of caper could serve as a source of numerous biochemical substances from industry and medicine [15].

Numerous scientific studies have been conducted on the bioecological and medicinal properties of *C. spinosa* L., chemical composition of its fruits, cultivation techniques, botanical properties [16], development of desert environments, application in biological regeneration, reproduction, and development of biotechnology [17].

Although the properties and characteristics of the soils of the Fergana Valley, where these plant areas are widespread, have been studied by Gulom Yuldashev with co-authors, the soil the biogeochemistry in the medicinal plant system has not been thoroughly studied [18,19]. According to [20,21] light gray soils (Calcisols) of the semi-desert region are mountainous and are included in the subtropical natural-climatic zone of Central Asia. These Calcisols were previously characterized in detail in terms of taxonomy, soil chemistry and texture [22]. It was emphasized that Calcisols play a crucial role in caper population existence in severe climatic conditions [23] The 15th East Fergana District is referred to as the Akbuyra-Aravan District [24]. Calcisols at an altitude of 550–750 m above sea level are formed on the proluvial rocks, consisting of fine-grained sandy and loamy rocks. At the same time, coarse, gravel-soft rocks and gravels are also selected, which are covered with fine-grained sandy and loamy rocks.

Cenopopulation analysis of the distribution of this plant species in the southern Fergana hills, biogeochemical properties of the plant and its distribution soils, elemental composition of soils and plant organs, medicinal properties, and raw material reserves are not sufficient. In addition, the study of biomorphological features of *C*. *spinosa* L., distribution areas in the Calcisols of southern Fergana, the study of the elemental composition of vegetative and generative organs, the location of this species in the vegetation, and its cenopopulation analysis.

In the following period, population growth dramatically increased the demand for medicinal plants. As a result, due to the unplanned use of medicinal plants, their natural resources are reduced, and even some species have to be considered as species with protected status. Therefore, the cenopopulation analysis, which reveals the soil-climatic conditions of each species and the laws of natural regeneration, is of great scientific and practical importance. Thus, the aim of this study was to investigate the dynamics of the cenopopulation of caper and geochemical peculiarities of the environments of caper in Fergana valley. Following objectives were formulated: (1) to evaluate the dynamics of cenopopulation within the onthogenesis of plants in one season, (2) to identify soil types, key soil properties and to characterize environments, suitable for plantation of *C.*
*spinosa* L.i in oases of South Uzbekistan and (3) to investigate biogeochemical features of soil and various parts of the *C.*
*spinosa* L. Plants.

## 2. Materials and Methods

Cenopopulation analysis of biomorphological properties of plants during ontogenesis was studied using the methods of O.V.Smirnova and others [25], root system P.K.Krasilnikov [26]. At the same time, the name of the plant species is based on the work “Determinant the plants of Central Asia” [27] and the International Electronic Database Names Index [28]. Research work has been carried out in the southern Fergana hills (Arsif, Satkak, Chimgan, Altiariq) since 2017. These hills are located at an altitude of 500–750 m above sea level. These areas are weak and moderately plastered, gravelly, skeletal, with perennial precipitation of 180–200 mm. The evaporation rate is estimated about 1200–1500 mm. Thus, the soils investigated are upward flow of moisture. These soils have low and low levels of humus and key nutrients. The level of coverage with natural vegetation is 40–60% depending on the slope exposure. The main parts of these plants are ephemeral and ephemeroids.

The object of study is the light Calcisols and irrigated soils (old irrigated hydromorphic soil (7 A) and recently irrigated hydromorphic soils (6 A) formed on the eroded weakly skeletal alluvial-proluvial rocks of southern Fergana, and the plant *C. spinosa* L., which is widespread in this area. Soil profile consists of the following horizons: A—humus accumulative with high roots content, BCA—horizon of secondary accumulation of carbonates, C—pebble gypsum parent materials. Detailed soil characteristics of this region are provided in [22]. In general, these soils are formed on clayely textured parent materials and demonstrate features of accumulation of secondary carbonates. Topsoil horizon is characterized by light gray color and crumby aggregate structure. 

The location of plots is given in Figure 1.

Phenological and morphological methods were used [29] methods of studying the seasonal development of the plants, i.e., the formation of grass during the growing season, the formation of true leaves, the growth of twigs and stems, budding, the beginning and end of flowering, the formation and ripening of fruits, the end of the growing season. The study of the duration of the flowering period was conducted in the identified 10 observational experimental populations.

The morphogenetic methods of V.V.Dokuchaev, pedogeochemical approaches of M.A. Glazovskaya, and A.I.Perelman were used as the main methods in the study of soil properties of *C. spinosa* L. [30]. Soil profiles were described and identified according to WRB 2015 [31].

The number of soil transects was 10, of which 8 represented automorphous Calcisols and 2 represented semi-hydromorphic soils. In addition, according to the methodology of the field survey around each reference soil section, samples were taken from four small sections for soil description and sampling. For chemical analysis 20 plants were selected and each plant was analyzed in 5 replicates. Roots, stems, leaves, buds, flowers, and fruits were sampled.

Mathematical processing was performed according to the method of Kuziev R.K., Yuldashev G. et al. [32,33]. Statistical analysis of cenopopulation dynamics varies in the limit that standard deviation (δ) ±1.0–±1.9, coefficient of variation (v) ±17.5–±43.7, mean error (m) ±0.3–±0.6, degree of reliability (t) 7.2–18.1 and degree of accuracy (P) varies from 5.5–13.8.

The neutron activation method was used for elemental analysis at the Institute of Nuclear Physics, ANRUz. The Microelement software was applied for statistical processing with the methodology by Kuziev et al. [33], Kuziev and Sektimenko [34]. Neutron activation analysis is one of the nuclear physical methods of elemental analysis, which is based on measuring the gamma activity of radionuclides formed in the analyzed sample as a result of nuclear reactions (activation) of isotopes of the elements being determined when irradiated with a thermal neutron flux. Identification of the generated radionuclides is carried out by the type of radiation, its energy, intensity, and half-life of radionuclides. For this, gamma spectrometers and high-resolution semiconductor detectors are usually used, less often spintillation detectors. Quantitative analysis is based on the fact that the activity of the resulting radionuclide is proportional to the amount of the element being determined in a wide range of concentrations. The relative method for calculating concentrations is applied using reference samples, also reference samples, where the required elements are known. The high penetrating power of radioactive radiation allows analysis without destroying the sample. Neutron activation analysis of soil samples was carried out at the BBP-CM nuclear reactor of the Institute of Nuclear Physics of the Academy of Sciences of the Republic of Uzbekistan. The neutron flux in the irradiation channels is 5 × 10^13^ neutron/cm^2^ s. When irradiated with a neutron flux, natural chemical elements present in the samples under study are converted into radioactive isotopes with different half-lives. It is convenient to measure the content of some elements by short-lived radioisotopes, some by medium-lived radioisotopes, and some by long-lived isotopes. Therefore, samples under study are usually irradiated in three modes: for analysis of short-lived, medium-lived, and long-lived radioisotopes. Short-lived ones have a half-life from a few seconds to several hours, medium-lived from several hours to several days, and long-lived have a half-life of months, years after irradiation in a nuclear reactor.

The fine earth samples were prepared by the grounding of the bulk soil mass and passing the soil through a 2 mm sieve. Soil chemical analyses were performed according to the routine methods of soil chemistry [35]: total carbon (Tyurin’s method), pH in water suspension (conductometric method.) Total nitrogen and phosphorous and potassium were determined according to the Maltseva and Grisenko method [36] 2 g of fine earth were put into a flask and amended by 5 g of H_2_SO_4_ and HClO_4_ mixture. After 30 min of reaction, the mixture was boiled on the electric stove. The mixture was poured into the measuring flask and passed through the filter. A 2 mm aliquot was taken and poured into a 50 mL flask and further amended by Rochelle’s salt. Then the mixture was neutralized by sodium hydroxide, the final solution was used for colorimetrical determination of the oxides with a coloring reagent (Nessler solution) 

## 3. Results and Discussion

A study of the *C. spinosa* L cenopopulation and its conditions of existence in the dry subtropics of southern Uzbekistan was carried out. Pictures of vegetative *C. spinosa* L. are giver on Figure 2. For the first time, the complex characteristic of soils of automorphous and semi-hydromorphous positions of relief was given. Soil properties and biogeochemical parameters of ecosystems were characterized. The connection between the chemical composition of soils and the biochemical composition of various parts of studied plants was revealed.

In biogeochemical research, the identification of correlations between the chemical, more precisely the elemental composition of plants and the elemental composition of the soils in which they grow is of great scientific and practical importance in establishing programmed yields in agriculture. In particular, knowledge of the exact amount of macro and micronutrients in the generative and vegetative organs of the plant *C. spinosa* L. roots, stems, leaves, buds, flowers, fruits also expands the use of this plant species in phytobars, food and pharmaceutical industries. 

The study of ontogenetic and phenological properties of plants is one of the most convenient and effective methods to determine changes in different phases of the observed plant species, their resistance to environmental conditions, productivity, as well as the rhythm of life processes in them. *C. spinosa* L. belongs to Capparaceae. Capparidaceae is a family of two genus plants with 40 genera and 850 species. Most of the plants belonging to the Capparis family are wild species, which are mainly distributed in arid regions of tropical and subtropical regions [8,37]. The Fergana Valley is a unique subtropical ecosystem where an ancient agricultural oasis was formed. In terms of climatic and geogenic characteristics, it is similar to the places to which the specified plant is native [23,38]. The natural distribution of *C. spinosa* L. in Uzbekistan depends on different geographical conditions [16]. The patterns of geographical distribution and ecological features of this plant have been essentially underinvestigated, both in Uzbekistan and throughout Central Asia [39].

During our studies, it was found that this species has entered the desert and semi-desert zone, in the foothills and lower mountain regions, sometimes up to the middle zone of the mountains. The study of the biological and ecological properties of any plant requires, first of all, the study of its condition under natural conditions. The natural adaptation of *C. spinosa* L. to soil and air drought allows it to grow in arid areas where water is scarce and in soils with high concentrations of water-soluble salts. Thus, environments of *C. spinosa* L. habitat were characterized in detail for Souht Uzbekistan for the first time.

*C. spinosa* L., has been observed by Saksali et al. as a promising plant that can grow in arid and strongly saline soils with nutrient deficiencies as well as in high-temperature regions [33]. While the soils of Uzbekistan are normally salinized, this species became very perspective for current agriculture, which is faced with water deficit in the condition of irrigation problems.

The length of the stem of plants distributed in the study areas reached 70–170 cm depending on the growing conditions. The inside of the newly formed young stems is covered with fine short hairs, but the hairs fall off as the branch grows during the growing season. The color of the stem is green, there are twisted spines on the underside of the leaf bundle. The number of side branches was 2–6, depending on the stage of development, and was 10–15 cm long. Poya diameter 7–12 mm. The leaves on the stem of the plant differ in shape, width, and length. Usually the leaf shape is round, inverted ovate or elliptical, 3–6 cm long, green, hairless or the lower side has scattered hairs, arranged in series on the main stem and lateral branches through a short leaf band.

The flowers are solitary, slightly zygomorphic, 5–8 cm in size, fragrant, located in the axils of one leaf, the petals are 4, curved, ovoid, green, covered with small short hairs on the outside. The petals are 4, but 2 are up to half, white or light pink, many paternal pollen, varying in length, pollinated, brown (flowers turn red after pollination). The flowers are 4–6 cm long. Normally, it blooms in April–May, depending on the amount of precipitation in the study area. The fruit is a multi-seeded berry. The color is green, with long white stripes. The shape is inverted ovate, oblong, walnut or round, many-seeded, or elongated. The outside is smooth, the inside is dark red. The fruit resembles the appearance of a watermelon. When the fruit was ripe, the fruit peel turned outwards and opened. Fruits are 3–5 cm long and 1.3–2.7 cm wide.

Nowadays, the regular addition of *C. spinosa* L. to the diet helps to relieve rheumatic pains. Currently, all parts of the plant are used in modern medicine and folk medicine in the treatment of meteorism, goiter, dentistry (gum and dental diseases), cardiovascular diseases, as well as hypertension, pruritus, jaundice, neurosis, brucellosis [5]. Biochemical activity of caper extracts could increases under various chemical treatments, this topic is quite promising and perspective and should be investigated in more detail [40,41]. 

Given the growing demand for raw materials of *C. spinosa* L., the need for in-depth study of its biogeochemistry and agroecology was put on the agenda, given the special attention paid to its export potential. It is important to determine the position of the *C. spinosa* L. in the vegetation cover, the status, ontogeny, and viability of the populations that determine its natural recovery, and thus its current and future raw material reserves.

Results of the analysis of *C. spinosa* L. cenopopulation in 10 experimental observation sites (100 m^2^ each) in Arsif, Satkak, Chimgan, and Altiariq hills were as follows (Table 1, Figure 3): grass (*p*) plants averaged 6.0, 5.0 plants belonging to the juvenile (*j*) state, 10.8 plants belonging to the immature (*im*) stage, 5.5 plants belonging to the virginil (*v*) state, 3.5 plants belonging to the generative (*g*) period, plants typical of the senile (s) period were 2.3.

On the territory of Arsif and Satkak landscapes *C. spinosa* L. is distributed on northern and western slopes; these slopes are relatively shady and less heated, so the soils in these areas have slightly higher humidity and lower air temperatures.

In the ecosystems of Chimyan and Altyaryk, the plant *C. spinosa* L. is predominantly distributed on eastern slopes, where sunlight falls relatively steeply on this area, and hence the heat is greater and the termic regime is different. Thus, we can conclude that the quality of the environments essentially affect cenopopulation dynamics within the ontogenesis of *C. spinosa* L.

In relatively warm climates, the growth of these plant grasses occurs in early April. Observations in our experimental fields revealed that *C. spinosa* L. grasses (Figure 2) germinate in late April to early May. In grasses, the seeds have two leaves, 2–3 cm in height, and the roots are 12–14 cm long and branches up to two rows. It was observed that 80–85% of the grasses pass to the juvenile stage in late May and early June.

In the juvenile mode, the seeds continue to grow in the palla leaves. Plants belonging to this stage are 5–7 cm tall, forming 3–4 leaves, the first true leaves are smaller. It was later observed that each chin leaf grows longer than the previous one. The main root reaches 20–22 cm and branches in 2–3 order. The peculiarity of this stage is explained by the drying of the seed palla leaves.

Plants belonging to the immature stage are observed in mid-June, their height is 15–20 cm, and the main root is 45–50 cm, branched to 2–3 (4) order. It was observed that 60–70% of plants belonging to the immature stage go to the virginil stage in late June and early July, and 10–15% go to the virginil stage in early May after the winter dormancy period. The duration of the immature phase lasts from 20–25 days to 10 months.

Seedlings of *C. spinosa* L., plants belonging to the juvenile and immature stages, are resistant to drought, but most of them die due to the crushing of livestock.

Plants belonging to the virginil stage are observed in late June to early July, the length of their main stem reaches 40–80 cm, and it branches up to 2 orders. The root reaches 90–110 cm and branches in 3–4 orders. At this stage is characterized by the formation of thorns on the stems and the thickening of the main root (diameter 4–5 mm). The duration of the virginil state depends in many respects on external environmental factors. It was observed that 15–20 per cent of virginil plants enter the full generative period in the first year and the rest in the second year.

Vegetation of plants belonging to the middle-aged generative stage in the Arsif hills lasted from April to December. The length of the main generative stem is 70–170 cm, branched to 2–3 rows, with leaves 4 × 3 cm. In one bush formed an average of 9–14 generative stems. The growing of generative plants was observed in May, flowering in late May, and the formation of fruits began in the second half of June.

The fruiting process of *C. spinosa* L. lasted from June to October. One bush, *C. spinosa* L., produced an average of more than 80 fruits (180–210 in the Arsif and Satkak hills) on the Chimgan hills, and 150 on some bushes. An average of 220–235 seeds were observed in each fruit, the absolute weight of 1000 seeds was 7.25 g. Seed length was about 1–3 mm [20] kidney-shaped, brown colored. It was found that the length of plant seeds in the study areas was 2.8–3.3 mm. Fruit ripening took place in the second decade of July in the hills of Arsif and Satkak, and in the hills of Chimgan and Altiyarik in late July and early August. 

A few works were published previously about ecology of *C. spinosa* L. in Uzbekistan [16,17]. However, the properties of the plant and the soil, its biogeochemical properties in relation to the composition of the chemical elements have not been studied. In our study, the cenopopulation of the *C. spinosa* L. plant, the migration of chemical elements in the plant-soil chain, and the biogeochemical properties were studied. The population studies of *C. spinosa* L. have been carried out in a neighbouring country, Iran. It was found that the present population of the plant is very heterogeneous at the genetic level, which is due, among other things, to environmental factors [42].

This plant is valued by many peoples of the world as a potential source of nutrients, vitamins, phenolic compounds, flavonoids, nutrients in its organs, as well as its strong antioxidant properties and ability to grow in arid conditions. According to the data [15], 100 g of *C. spinosa* L. contains: phosphorus (679 mg/g), sodium (652 mg/g), calcium (419 mg/g), magnesium (213 mg/g), potassium (157 mg/g), macro and micronutrients such as iron (6.8 mg/g), zinc (5.5 mg/g), manganese (3.30 mg/g) have been reported [15].

*C. spinosa* L. buds contain elements of vitamin K, potassium, calcium and magnesium, which strengthen the bones and prevent the development of osteoporosis. *C. spinosa* L. prevents hair loss due to the presence of iron and B vitamins, and makes hair grow beautiful and shiny [43]. The recent discovery of the substance stachidrin in *C. spinosa* L. has aroused great interest among scientists. This substance has strong antimetastatic properties and is used in the treatment of prostate cancer. This substance stops the growth and development of cancer cells. This scientific breakthrough is important in the development of anti-cancer drugs. The use of this plant in bowel cleansing and prevention of colon cancer plays an important role [44].

The high content of sodium in the vegetative and generative organs of plant species, especially in salinized soils of desert areas, requires caution in some diseases. *C. spinosa* L. may be problematic in humans when consumed in the existence of the following diseases: hypotension, constipation, and is not recommended for pregnant women because high sodium levels have been shown to affect the fetus, in individual cases, and in some cases to cause allergies [17]. Although biologically active organic substances in medicinal plants have been systematically studied, biologically active mineral elements have not been adequately studied. As a result of the growing number of drugs made from medicinal plants, the analysis of their elemental composition [45,46] and biochemical and physichochemical properties [47].

The sodium content in the genetic layers of Calcisols ranges from 0.68–0.94 μg/g, which indicates its accumulation in 1.08–1.49 times higher than in Phaeozems of subboreal steppe ecosystem. The medicinal plant C. spinosa L. absorbs in its organs an average of 439 μg/g sodium (root skin—1300 μg/g, root core—1200 μg/g, stem—75 μg/g, leaf—84 μg/g, bud—135 μg/g, flowers—130 μg/g, fruit—151 μg/g).

The granulometric composition of the soil investigated is characterized by the amount of sand, silt and clay particles in the fine earth. The amount of each particle, together with the amount of other particles, affects the number of properties of the soil. In particular, small particles allow the particles of soil to stick together and grow, and participate in the formation of structural aggregates in the soil. Larger particles are involved in the formation of soil skeletons, improving water and air permeability. Data obtained on soil texture vertical distribution (Table 2) are in good correspondence with recently published ones [22]. The studied soils have mainly light and medium sandy mechanical compositions. These soils are rich in a number of macro- and micronutrients (Table 2). In addition, the mechanical composition of the soil is closely related to the migration of its elements. Due to weak development of the humus horizon Calcisols of Central Fergana are faced with wind and water erosion. The presence of caper on the soil surface could play an anti-erosion soil protective role, which has been reported recently [48,49].

It is known that humus is a complex structural substance with a variable composition, the composition of which is constantly changing and renewing. This dynamic condition applies primarily to humic substances, such as humic and ulminic acids, fulvic acids, hyimatomelane acid, etc., along with humic substances, carbohydrates, organic acids, alcohols, hydrocarbons, ethers, aldehydes, nitrogenous substances, and others. Calcisols investigated do not show high content of organic carbon, which well corresponds with data, published recently [22]. The dynamics of soil formation take place through the influence of different levels of organic matter on itself, on the parent rocks, and this process is considered biogeochemical. Organic matter, in whatever quantity and quality it is in the soil, plays the role of a source of carbon dioxide, nutrients, and energy for plants. Humus increases the stability of agriculture in the soil and plays a key role in soil formation. Its optimal amount of humus in the soil regulates the thermal regime, creates a valuable structure, and acts as an energy reserve. Very low levels of C/N ratios are typical for soils investigated. This indicates a very intensive soil organic matter mineralization rate and high rate of soil biological processes. 

In the soils of Central Fergana, the root system of natural and cultivated plants is located in the very superficial layers of the soil, so the humus layer is weak in these soils. This is true even for soils with arzyk-shokh and shokh-arzyk horizons at different depths (Table 3).

Based on the data, given in the Table 3, the amount of humus in the drive and subsoil layers of the old irrigated (7A) soil sections fluctuates around 1.1–1.5%. The amount of humus in the subsequent layers of these soils is also relatively high.

In the upper driving layers of newly irrigated soils, humus content is about 0.8–0.9%.

The total amount of nitrogen in the soil humus is correlated, so the law of cross-sectional changes in the amount of nitrogen in the studied soils is reminiscent of humus. The C:N ratio in the soil is a relative measure of the nitrogen content of humus, a characteristic of most irrigated soils. This ratio Kuziyev R.K. data has been found to vary from 7.2 to 13.5, especially in gray-oasis soils. For gray soils, this value is 8 and indicates that nitrogen is rich in nitrogen [50]. The C:N ratio in the surface, shallow, deep-layered soils of the soil we studied is 5.2–7.9. In current research we classified the soil investigated as Calcisols, which in [22] were identified as gray-oasis soils. Key soil chemical and granulometric (texture) properties are comparable with those published previously [22]. 

The total amount of phosphorus and potassium in these soils is not high. For example, gross phosphorus fluctuates around 0.03–0.125% in soil sections. The amount of total potassium fluctuates around 0.80–2.45%, of course 0.80–0.90% corresponds to the man-made strata, while 2.45% corresponds to the driving layer of old irrigated soils. According to the amount of mobile nutrients, these soils belong to the low-income group.

In this regard, we can conclude that the total phosphorus and potassium content in the soils of the old irrigated deep 7A section (93–111 cm) and the newly irrigated hydromorphic 6A section (32–55 cm) also decreased sharply as humus.

It is known that the amount of chemical elements and substances varies in different soil types [30] (Table 4). Therefore, the chemical element composition of plants depends on the amount of chemical elements in the soil in which the plant grows. Recently is was shown that biomass production of caper essentially depends on the uptake of P, K^+^, Mg^2+^, Fe^2+^ and Zn^2+^ which could be problematic in desert and dry conditions. The regulation of uptake is possible by adding native arbuscular mycorrhizal fungi inoculum into soil under plantation of capper. In any case numerous eco-engeniring strategies could be elaborated with the aim to increase the productiveness and yield of *C. spinosa* [51]. 

According to the data in Table 2 above, the amount of chemical elements in soils formed under different conditions varies, due to soil genesis, soil-climatic conditions, use in agriculture, and so on. For example, Mn and Zn were found to be higher in the 32–55 cm layer of newly irrigated hydromorphic soils than in other soil layers, while Molibden in Calcisols were found to be present in the lowest amounts in the 0–10 cm layer. The content of microelements in Calcisol is essentially lower than in other soils investigated. The first reason is the high content of carbonates, which plays an essential role in the binding of microelements in soils. The second reason is that irrigated soil characterizes by higher solubility of any elements and due to this fact, the concentration of available microelements in these soils is higher.

Strengthening the process of hydromorphism leads to an increase in the amount of Mn, Zn, Mo in the soil and its layers from Calcisols to meadow soils. This can also be explained by the fact that the geochemical migration of these elements towards a naturally dependent landscape is accumulated with taking into account that fact that investigated soil were not amended be any fertilizers.

The elemental composition of the *C. spinosa* L. which is widespread in the Calcisols of South Fergana, changes under the influence of soil properties, plant type, natural climatic conditions and other factors. It was observed that the amount of elements in the composition of *C. spinosa* L. varies several thousand times depending on the physiological properties of plant organs.

Table 5 shows that the amount of micronutrients studied varies in plant organs, or Mn 9–100 μg/g, Mo-0.29–5.2 μg/g, Co-0.086–0.25 μg/g and Zn-5 μg/g. Oscillations in the range of 1–34.1 μg/g were detected. Of the trace elements studied for absorption into plant organs, the highest amount is Mo (52 μg/g in the root bark) and the lowest amount is Co (0.018 μg/g in the stem). Note that the element Mn is absorbed in very high amounts in the plant leaf, Mo in the root bark compared to other organs, and Zn is accumulated in large quantities in the fruit. If we pay attention to the classification of the studied elements in terms of their biological role [52], they are among the biogenic, essential elements necessary for life.

Our data shows that Mo belongs to the group of strong and very strong aggregates by root bark and fruit. According to the range of biological absorption coefficients, the elements Co, Zn, and Mn belong to the group of very weak, weak, and moderately biodegradable, respectively.

There is a positive correlation between the content of trace elements in the soil and their amount in the organs of medicinal plants. The range of correlation is from 0.80 to 0.95 (Table 6). The highest levels of correlation were for levels of cobalt, indicating its maximum mobility in the soil-plant system.

This, in turn, satisfies the need for certain macro and micronutrients by consuming a biologically active supplement made from this medicinal plant as well as the daily norm of food. The study of the correlation between the elemental composition of the plant and the elemental composition of the soil in which it grows expands the scope of its use in folk medicine, phytobars, modern medicine, and the pharmaceutical industry. This makes a huge contribution to socio-economic development.

The main economic importance of *C. spinosa* L. is related to the types of products made from it. In particular, pickled flower buds, known as “capers” or “caper berry”, are the main subject of trade in international markets. In recent years, the annual growth rate of production from *C. spinosa* L. has increased by 6%. Currently, pumpkin is valued as an important consumer product in the United States and about 60 countries around the world, where the cost of 1 kg of ready-to-eat product is $25. From this round, the Chinese earn three million a year. They are making a profit in the amount of USD. Today, the Kingdom of Saudi Arabia, Lebanon, Syria, and the Mediterranean countries have proposed *C. spinosa* L. as the main crop type to raise the socio-economic level [10]. However, some cosmetic products derived from the fruit extract of *C. spinosa* L. (e.g., Gatuline Derma-Sensitive-$74.99; Skin moon-$76; Skin save-$7.70) are used as anti-aging, skin protection, or anti-inflammatory agents. was commercialized and put up for sale [41].

In the Jizzakh region of the country, 12 enterprises have launched the export of *C. spinosa* L. It should be noted that until recent years, this plant has been neglected, whereas *C. spinosa* L. is a very valuable raw material in the pharmaceutical, and food industry. In 2019, in the Jizzakh region, this plant was harvested from existing natural resources and cultivated. 1909 tons of ready-to-eat products were made from its flower buds and fruits. It exported $3.2 million worth of goods to Turkey and Spain. In addition to productivity and yield parameters *C. spinosa* L. this plant species is characterized by good adaptation to conditions of marginal environments [53] and, thus, could be used for land reclamation and stabilization of unstable soil surfaces. 

## 4. Conclusions

Soil-plant interaction in a case study of *C. spinosa* L. plantation located on dry subtropical Calcisols of Central Fergana provided for the first time for Central Asia on the territory of the Former Soviet Union (NIS). It was shown, that *C. spinosa* L. is well adapted to dry soils with alkaline reaction, low humus percentages, and middle key nutrients content. The increasing of soils humidity is not a critical factor for growing *C. spinosa* L. dynamics. 

Cenopopulational observation of *C. spinosa* L. in a field experiment in case of the various soils was conducted. According to the observations, the viability, and drought tolerance of *C. spinosa* L. populations is relatively high, and studied plants of the population belonging to the immature and virginil stage are 5–10 times more than senile. This shows that it is possible to collect raw materials of the caper from the southern Fergana hills with severe climatic conditions on a regular basis. Determining the quantitative supply of nutrients and medicinal substances in *C. spinosa* L. and other medicinal plants will further increase the productivity and increase the medical features of vegetated medical plants in the Central Fergana oasis.

The series of biological absorption is assimilated in the form of *0, 0n* → *0,*
*n* → *n* → *10n*, depending on the amount of elements studied. Molybdenum showed features of very strong biological accumulation in plants, while manganese, Zinc, and Cobalt occupy a medium, weak, and very weak retention line.

By studying the amount of chemical elements in the organs of the plant *C. spinosa* L. depending on the composition of the soil, it is possible to assess its sanitary and hygienic characteristics, as well as the level of safety in pharmaceutical use and food quality. Because this plant has anti-cancer, anti-microbial, and anti-viral effects, further surveys of its chemical composition and biochemical specificity should be provided. *C. spinosa* L. can be used as a raw material in the creation of new medicines.

The intensively increasing demand for natural and environmentally friendly products made from *C. spinosa* L. the global increase in its use in the food industry, and modern medicine, the increase in natural products made from it in the pharmaceutical and cosmetic industries, in turn, increase the pressure on natural resources of this plant species. This would jeopardize the natural reserves of *C. spinosa* L. in the nearest future. Thus, further research of *C. spinosa* L. population dynamics in various environmental conditions should be conducted together with a simultaneous determination of the productivity of the plant species.

## Figures and Tables

**Figure 1 plants-11-01628-f001:**
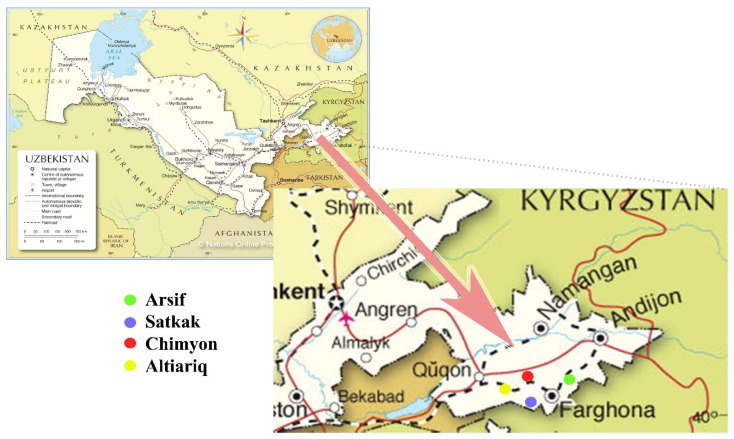
Map of research area, Fergana valley (Source: https://www.nationsonline.org/oneworld/map/uzbekistan-political-map.htm (accessed on 5 May 2022).

**Figure 2 plants-11-01628-f002:**
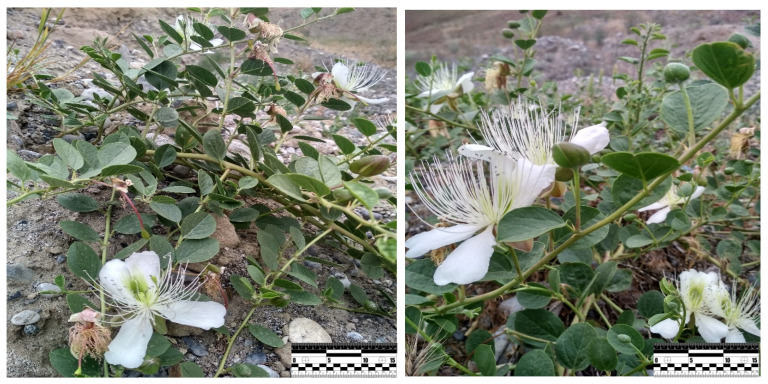
The plant *C. spinosa* L. growing in the experimental area (Scale: 1:50).

**Figure 3 plants-11-01628-f003:**
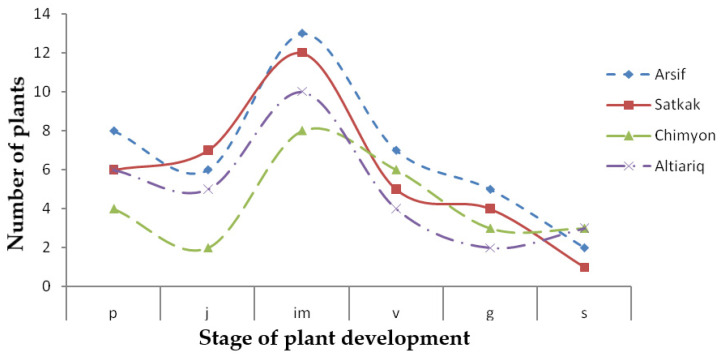
Graph of *C. spinosa* L. cenopopulation in elementary landscapes in colored curved lines. *x*-axis—*p*—grass, *j*—juvenile, *im*—immature, *v*—virginil, g—generative, *s*—senile, *y*-axis—number of plants, units per 10 m^2^, in each stage experiments.

**Table 1 plants-11-01628-t001:** Cenopopulation of *C. spinosa* L.

Elementary Landscape	Number of Individuals, Pieces (*n* = 10)						Total
	*p*	*j*	*im*	*v*	*g*	*s*	
Arsif	8	6	13	7	5	2	41
Satkak	6	7	12	5	4	1	35
Chimyon	4	2	8	6	3	3	26
Altiariq	6	5	10	4	2	3	30
Average	6	5	10.8	5.5	3.5	2.3	33

**Table 2 plants-11-01628-t002:** Soil particle size distribution in the fine earth.

Depth, cm	Particle Size Fractions, mm							<0.01
	>0.25	0.25–0.1	0.1–0.05	0.05–0.01	0.01–0.005	0.005–0.001	<0.001	
Calcisol								
0–10	5.8	2.7	27.8	33.5	11.7	9.6	8.9	30.2
10–30	7.2	2.6	25.9	34.3	10.1	9.5	10.4	30
30–56	5.3	3.9	25.6	36.6	11.4	7.7	9.5	28.6
56–120	8.2	6.7	29.5	36.3	9.6	4.3	5.4	19.3
Old irrigated hydromorphic soil								
0–28	3.52	33.8	21.43	8.4	8.26	9.39	15.2	32.85
28–36	3.6	21.47	19.2	30.77	8.2	5.36	11.4	24.96
36–93	21.4	15.49	18.1	17.11	7.1	13.1	7.7	27.9
111–140	11.3	15.2	18.1	15.2	16.9	12.2	11.1	40.2
140–180	10.27	13.1	25.2	11.33	12.8	13.1	14.2	40.1
Recently irrigated hydromorphic soil								
0–18	4.02	27.8	19.1	27.11	6.35	5.46	10.16	21.97
18–32	6.71	19.4	17.2	32.37	5.26	11.26	7.8	24.32
55–80	22.3	16.1	19.21	18.58	6.14	10.31	7.36	23.81
80–140	12.1	14.2	15.5	30.7	6.2	10.2	11.1	27.5
140–200	11.3	14.1	26.2	2.04	14.66	15.8	15.9	46.36

**Table 3 plants-11-01628-t003:** Agrochemical characteristics of soils.

Depth, [cm]	TOC, %	C:N	Bulk Elements Content, %			Mobile, mg/kg		
			N	P_2_O_5_	K_2_O	N	P_2_O_5_	K_2_O
Calcisol								
0–10	1.65 ± 0.07	9.3	0.12	0.133	2.11	12.1	21.3	220
10–30	0.72 ± 0.05	6.9	0.07	0.121	1.76	8.9	11.7	199.8
30–56	0.55 ± 0.04	6.5	0.06	0.105	1.23	6.7	5.6	110
56–120	0.31 ± 0.02	5.7	0.04	0.085	0.79	5.9	-	97.5
Old irrigated hydromorphic soil								
0–28	1.23 ± 0.02	6	0.14	0.125	2.35	18.9	22.85	195
28–36	1.12 ± 0.03	5.8	0.13	0.127	2.17	7.8	16.25	110
36–93	0.42 ± 0.04	4.9	0.06	0.11	1.85	-	-	-
93–111	-	-	-	0.2	0.9	-	-	-
111–140	0.32 ± 0.04	6.8	0.03	0.04	1.55	-	-	-
140–180	0.21 ± 0.02	5.6	0.03	0.03	1.55	-	-	-
Recently irrigated hydromorphic soil								
0–18	0.93 ± 0.05	5.5	0.11	0.103	2.1	17.5	18.25	140
18–32	0.72 ± 0.05	5.2	0.089	0.1	1.95	8.5	15.1	85
32–55	-	-	-	0.025	0.8	-	-	-
55–80	0.28 ± 0.04	5.3	0.035	0.105	1.45	-	-	-
80–140	0.22 ± 0.02	6.7	0.021	0.04	1.55	-	-	-
140–200	0.22 ± 0.03	7.3	0.019	0.035	1.5	-	-	-

**Table 4 plants-11-01628-t004:** Amount of microelements in soils (*n* = 7).

Depth of Sample, cm	Element (μg/g)			
	Mn	Zn	Co	Mo
Calcisol				
0–10	370 ± 12	55.8 ± 3.5	4.71 ± 0.31	<0.10
10–30	430 ± 11	37.1 ± 2.7	7.35 ± 0.43	1.60 ± 0.04
30–56	520 ± 15	59.8 ± 4.5	7.45 ± 0.45	0.55 ± 0.05
56–120	420 ± 11	61.1 ± 4.9	8.58 ± 0.65	1.10 ± 0.07
Old irrigated hydromorphic soil				
0–28	710 ± 14	85.1 ± 5.6	2.7 ± 0.13	2.65 ± 0.09
28–36	620 ± 16	68.1 ± 4.5	3.0 ± 0.12	2.35 ± 0.08
36–93	530 ± 12	71.2 ± 3.4	2.9 ± 0.12	2.50 ± 0.06
93–111	830 ± 17	91.9 ± 7.8	5.4 ± 0.31	2.55 ± 0.07
111–140	620 ± 13	85.0 ± 5.6	6.6 ± 0.21	5.45 ± 0.17
140–200	650 ± 15	11.9 ± 2.3	9.0 ± 0.43	6.45 ± 0.13
Recently irrigated hydromorphic soil				
0–18	665 ± 16	77.0 ± 4.5	7.8 ± 0.22	2.60 ± 0.06
18–32	610 ± 14	61.0 ± 2.4	7.1 ± 0.25	2.55 ± 0.09
32–55	920 ± 20	98.1 ± 5.4	8.0 ± 0.18	2.60 ± 0.12
55–80	630 ± 8	86.0 ± 2.2	4.9 ± 0.21	4.35 ± 0.14
80–140	630 ± 11	80.2 ± 2.1	5.4 ± 0.14	4.20 ± 0.09
140–200	635 ± 9	120.0 ± 9.2	6.7 ± 0.16	6.90 ± 0.08

**Table 5 plants-11-01628-t005:** The amount of micronutrients in the organs of *C. spinosa* L. (μg/g) and the coefficient of biological absorption, (*n* = 14).

Plant Organ	Microelements (μg/g)				Biological Absorption Coefficient			
	Mn	Mo	Co	Zn	Mn	Mo	Co	Zn
Root skin	32 ± 3	5.20 ± 0.72	0.25 ± 0.03	27.0 ± 2.2	0.086	52.0	0.053	0.48
Root core	9.0 ± 1	0.55 ± 0.21	0.16 ± 0.02	5.10 ± 1.1	0.024	5.5	0.034	0.09
Stem	16 ± 2	0.29 ± 0.03	0.09 ± 0.02	14.00 ± 1.5	0.043	2.9	0.018	0.25
Leaf	100 ± 6	1.80 ± 0.07	0.18 ± 0.03	30.0 ± 2.0	0.270	1.8	0.038	0.54
Bud	26 ± 2	0.58 ± 0.03	0.12 ± 0.02	33.00 ± 2.3	0.070	5.8	0.025	0.59
Flowers	24 ± 2	0.50 ± 0.04	0.13 ± 0.02	30.00 ± 3.4	0.065	5.0	0.028	0.54
Fruit	34 ± 3	2.10 ± 0.09	0.19 ± 0.03	34.10 ± 3.1	0.092	21.00	0.040	0.61

**Table 6 plants-11-01628-t006:** Statistical treatment of soil and plants chemical analyses (n-7).

Pearson (r) Correlation Coefficient	Standard Error	Fisher’s Criterion
Manganesse		
0.80	0.5	1.13
Molibdene		
0.89	0.5	1.44
Cobalt		
0.95	0.5	1.96
Zink		
0.86	0.5	1.29

## Data Availability

Not applicable.

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
