# Peer review of "Capparis spinosa L. Cenopopulation and Biogeochemistry in South Uzbekistan"

_plants, 2022, doi:10.3390/plants11131628_

Round 1

Reviewer 1 Report

This manuscript contains sufficient novelty to be accepted for publication, but still minor modifications and suggestions are recommended to improve the quality.

 All minor remarks are highlighted in the manuscript.

·       Authors should avoid the lumping of references in the paper, but each should be discussed.

·       Figures are not well numbered in the manuscript.

·  In Figure 2, indicate what is abscissa and ordinate. Give full names of abbreviations (p, j, im…).

·       In tables 1, 3, 4, and 5 there should be a full stop instead of a comma.

Author Response

Dear reviewer!

Thank you so much for your comments and advices!

Comments and Suggestions for Authors

This manuscript contains sufficient novelty to be accepted for publication, but still minor modifications and suggestions are recommended to improve the quality.

All minor remarks are highlighted in the manuscript – reply: all comments and suggestions have been taken into account, text corrected.

Authors should avoid the lumping of references in the paper, but each should be discussed. reply– done.

Figures are not well numbered in the manuscript reply -corrected

In Figure 2, indicate what is abscissa and ordinate. Give full names of abbreviations (p, j, im…). reply - done

In tables 1, 3, 4, and 5 there should be a full stop instead of a comma  - reply –corrected

With kindest regards,

Evgeny Abakumov, corresponding author, Saint-Petersburg State University.

Reviewer 2 Report

The materials and methods section needs to be elaborated in a scientific and reproducible manner. There are many details missing from it.

The discussion section also needs Improvement. The authors should quote more recent literature while discussing their results.

The specific conclusion of this study is not quite clear.

Author Response

Dear reviewer!

Thank you so much for your comments and advices!

all comments and suggestions have been taken into account

The materials and methods section needs to be elaborated in a scientific and reproducible manner. There are many details missing from it. This chapter was essentially improved and amended.

The discussion section also needs Improvement. The authors should quote more recent literature while discussing their results. More references of more recent literature was added.

The specific conclusion of this study is not quite clear. The conclusion chapter was improved

With kindest regards,

Evgeny Abakumov, corresponding author, Saint-Petersburg State University.

Reviewer 3 Report

Review comments to the author

Title: ''Cenopopulation and biogeochemistry of Capparis spinosa L.''.

Manuscript ID: plants-1758128.

Introduction:

1- Page 1, Line 36: Add one space after the citation [1].

2- Page 2, Line 59: Add one space after the citation [5].

3- Page 2, Line 62: Add a dot (.) after the citations [6-8] instead of a comma (,).

4- Page 2, Line 65: Add one space after the citation [9].

5- Page 2, Line 75: Add one space between the word ''its'' and the word ''fruit''.

6- Page 2, Line 81: Add one space after the citations [16-23].

7- Page 2, Line 85: In the term ''15th'' the letters ''th'' should be typed in superscript font.

8- Page 2, Line 96: Add the word ''and'' before the section '', its cenopopulation analysis''.  

2. Materials and Methods

1- Page 4, Line 149: Add one space before the word ''Neutron''.

3. Results

1- This section should be written as ''Results and Discussion''.

2- Page 5, Line 192: The citation ''M.S.Saksali et al'' should be written as ''Saksali et al''.

3- Page 6, Line 236: ''Figure 1'' should be replaced by ''Figure 2''.

4- Page 7, Line 260: ''Figure 1'' should be replaced by ''Figure 2''.

5- Page 8, Line 281: ''Figure 1'' should be replaced by ''Figure 3'' and should be cited in the text.

Abbreviations:

- List of abbreviations should be inserted by the end of the manuscript before references.

References

1- All scientific names of plants and species should be written in italic fonts.

2- Journal names should be written in a uniform manner, either complete or abbreviated.

Author Response

Dear reviewer 3!

Dear reviewer!

Thank you so much for your comments and advices!

all comments and suggestions have been taken into account

Manuscript ID: plants-1758128.

Introduction:

1- Page 1, Line 36: Add one space after the citation [1]. - done

2- Page 2, Line 59: Add one space after the citation [5].- done

3- Page 2, Line 62: Add a dot (.) after the citations [6-8] instead of a comma (,). - done

4- Page 2, Line 65: Add one space after the citation [9]. - done

5- Page 2, Line 75: Add one space between the word ''its'' and the word ''fruit''. - done

6- Page 2, Line 81: Add one space after the citations [16-23].- done

7- Page 2, Line 85: In the term ''15th'' the letters ''th'' should be typed in superscript font. done

8- Page 2, Line 96: Add the word ''and'' before the section '', its cenopopulation analysis''   done

  1. Materials and Methods

1- Page 4, Line 149: Add one space before the word ''Neutron''. done

  1. Results

1- This section should be written as ''Results and Discussion' - done'.

2- Page 5, Line 192: The citation ''M.S.Saksali et al'' should be written as ''Saksali et al'' - done

3- Page 6, Line 236: ''Figure 1'' should be replaced by ''Figure 2''. done

4- Page 7, Line 260: ''Figure 1'' should be replaced by ''Figure 2''. done

5- Page 8, Line 281: ''Figure 1'' should be replaced by ''Figure 3'' and should be cited in the text. done

Abbreviations:

- List of abbreviations should be inserted by the end of the manuscript before references. done

References

1- All scientific names of plants and species should be written in italic fonts. - done

2- Journal names should be written in a uniform manner, either complete or abbreviated.-done

With kindest regards,

Evgeny Abakumov, corresponding author, Saint-Petersburg State University.

Round 2

Reviewer 2 Report

The authors have now improved their manuscript as per my suggestions, hence I recommend it for publication.

Author Response

Dear reviewer!

Thank you for your valuable comments, all of them were taken into account,

The revised manuscript still lacks a clear direction of its objectives. The authors have done work measuring plant traits and soil parameters at 10 investigated points but failed to clearly present their findings in the present paper. I have written up some advice and suggestions that can help improve the merit of the paper, however, the authors should decide if this is a scientific research paper or a descriptive paper of the plant across the Fergana Valley. Either way, the findings and their relevance to current knowledge of the topic must be clearly presented, and in accordance with the type of the paper, the title should be changed to better reflect.

-paper title has been changed.

-the objectives of study has be reformulated.

General comments:
The manuscript should be carefully checked for typos, there are many in the paper. E.g. Lines 61, 62, 80, 88, 91, 92, 134, 213, 316, 508.
G. Yuldashev, M.Isagaliev, S.Isaev, A.Turdaliev, etc. There is no need to abbreviate the first names of the authors. Please erase it throughout the paper.

  • corrected

Abstract:
Please provide a sentence or two of factual results from your study – has been added.
Line 24. The location of Fergana (e.g. Central Asia) also should be included – corrected.
Line 42. Please provide a reference – corrected.
Line 44. Which country? – corrected.
Line 46. Reference is missing – corrected.

Introduction:
Line 107. What is Red Book? – corrected.

Materials and Methods:
This section should also contain the exact numbers of soil and plant sampling that was done during the study. How often samples were taken? At what depth for soils? All 10 parcels? – information is added.
How many roots, stems, leaves, buds, flowers, and fruit samples were taken? From how many sampling points? – text is amended by this information
Statistical analyses also missing. Please amend this section with statistics that were used to compare the study sites, or as the paper is focused on: the dynamics of the cenopopulation. - сorrected
Line 135. It is not necessary to write who used this method, the main point the authors should make that they were using this method in their study. – the paragraph is reformulated
Figure 1. The map should be amended with sampling points for both soil and plants – amended.
Lines 149-150. This sentence is a repetition, consider removing it – corrected.

Results and Discussion:
This section is supposed to be starting with the results of the study. Please write down what were the results of the analyses here, and later discuss their meaning.
Lines 200-207. This paragraph belongs to the Introduction, not the Discussion section.-corrected
Lines 208-213. This paragraph does not have connection to the results, please amend. -corrected
Lines 244-254. This paragraph, again, is not a discussion, it is a description of the plants, but without a scientific base. The paragraph should be amended with statistical data collected from the different study points. -corrected
Line 266. Why this result is preliminary? When it comes to a publication, all analyses should be correctly done and documented – corrected.
Table 1. What are the differences between the landscapes? Why the results are not compared to each other? How have the different soil conditions affected the plant growth at the different phenological phases? – the comparison has been added.
Figure 2. A scale should have been added to the pictures – added.
Figure 3. Add Axis title for the y-axis, also some units, it is not clear what is being presented here – added.
Lines 318-326. This paragraph, again, does not provide with a discussion, it is an Introduction material with many repetitions to earlier parts of the paper – the sentence is reorganized.
Lines 337-346. This paragraph has no new information and was already described above. Please move this up where it is described and remove all repetitions – corrected.
Lines 347-355. Authors first should describe their findings and after discuss their meaning compared to international literature. What were the K contents of the selected plants at teh different sites? – sorry, we don’t measured K content in plants due to technical reasons, some discussion of has been added.
Lines 356-365. Again, what were the Sodium levels in the studied plants and soils? What were the differences between sites? Your results should be in the focus, not others – data about sodium content has been added.
Table 2. What do 1/OM, 7A, and 6A mean? Why not used all 10 experimental site data? – numbers of soil sections are deleted, soil names are given. Here we given the most representative types of soil typical for studied sites. This is necessary for primary description of soil of this part of Fergana valeey and will be interesting for broad audience of journal, while if we give all 10 plots, we have to delete all data of vertical soil profile.
Lines 442-443. And how different these sites were? Is it significant? Please provide some statistical data here. – comments on differences has been added, but statistic is given in the end off chapter.
Line 448. So these experimental sites are fertilized? – corrected.
Table 5. Why these data are not compared to the soil parameters? Earlier the soil data is divided by, it would greatly add merit to this paper if the plants were separately investigated by its soil types. – lower, additional table № 6 added with discussion of correlation parameters.
Tables in general: SD should be added, also some statistical information is necessary to understand what these numbers are meaning - corrected
Line 465. Please refer to Table or Figure. – has been added.
Lines 471-472. Yes, this is what I am still hoping to see in this paper, however there is no correlation done by the authors. I suggest some Pearson's numbers and significance to add meaning to these data. –correlation data has been added.
Lines 475-483. This paragraph, again, is not a discussion. It is not related to the data gathered by the authors. These are Introductory statements without connection between the authors' findings – has been deleted.
Lines 487-492. This paragraph is out of place, I suggest erasing it. corrected

Thank you,

With kindest regards,

Evgeny Abakumov, Saint-Petersburg State University